# Superplastic Deformation and Dynamic Recrystallization of a Novel Disc Superalloy GH4151

**DOI:** 10.3390/ma12223667

**Published:** 2019-11-07

**Authors:** Shaomin Lv, Chonglin Jia, Xinbo He, Zhipeng Wan, Xinxu Li, Xuanhui Qu

**Affiliations:** 1Institute for Advanced Materials and Technology, University of Science and Technology Beijing, Beijing 100083, China; lsmleon@163.com (S.L.); quxh@ustb.edu.cn (X.Q.); 2Science and Technology on Advanced High Temperature Structural Materials Laboratory, Beijing Institute of Aeronautical Materials, Beijing 100094, China; waynedapeng@163.com (Z.W.); lxx20110180@163.com (X.L.)

**Keywords:** GH4151 alloy, superplastic deformation, activation energy, dynamic recrystallization

## Abstract

The superplastic deformation of a hot-extruded GH4151 billet was investigated by means of tensile tests with the strain rates of 10^−4^ s^−1^, 5 × 10^−4^ s^−1^ and 10^−3^ s^−1^ and at temperatures at 1060 °C, 1080 °C and 1100 °C. The superplastic deformation of the GH4151 alloy was reported here for the first time. The results reveal that the uniform fine-grained GH4151 alloy exhibited an excellent superplasticity and high strain rate sensitivity (exceeded 0.5) under all experimental conditions. It was found that the increase of strain rate resulted in an increased average activation energy for superplastic deformation. A maximum elongation of 760.4% was determined at a temperature of 1080 °C and strain rate of 10^−3^ s^−1^. The average activation energy under different conditions suggested that the superplastic deformation with 1 × 10^−4^ s^−1^ in this experiment is mainly deemed as the grain boundary sliding controlled by grain boundary diffusion. However, with a higher stain rate of 5 × 10^−4^ s^−1^ and 1 × 10^−3^ s^−1^, the superplastic deformation is considered to be grain boundary sliding controlled by lattice diffusion. Based on the systematically microstructural examination using optical microscope (OM), SEM, electron backscatter diffraction (EBSD) and TEM techniques, the failure and dynamic recrystallization (DRX) nucleation mechanisms were proposed. The dominant nucleation mechanism of dynamic recrystallization (DRX) is the bulging of original grain boundaries, which is the typical feature of discontinuous dynamic recrystallization (DDRX), and continuous dynamic recrystallization (CDRX) is merely an assistant mechanism of DRX. The main contributions of DRX on superplasticity elongation were derived from its grain refinement process.

## 1. Introduction

Superplasticity is the capacity of a polycrystalline material to exhibit extremely high tensile ductility prior to failure [1,2]. The initial microstructure of fine equiaxed grains of size 10 μm or less was considered as one of the prerequisites for superplastic deformation [3]. Superplastic forming is a novel deformation technique for fabrication of varied and complicated shaped components due to utilizing low force, large plastic strains and few energy costs [4]. Accordingly, numerous investigations on superplastic deformation of materials have been studied, such as aluminum [5,6,7], titanium [8,9,10], magnesium alloy [11,12,13], high-entropy alloy [14,15] and nickel-base superalloys [16,17,18,19]. Specific mechanisms to superplastic behavior are: (i) grain boundary diffusion, (ii) grain rotation and rearrangement, (iii) grain boundary sliding and slip accommodation and finally migration of the grain boundaries [20].

For the investigation of superplasticity on nickel-based superalloys, Kaibyshev O.A. discussed the morphology of the gamma prime (γ′) during superplastic deformation. The influence factors of ductility and superplasticity of superalloy were analyzed, and the principle of recovering and improving high-temperature properties is put forward [21]. Superplasticity is a common phenomenon for nickel-based superalloys, which occurs at relatively high temperatures. Valitov et al. [22] reported the generation of micro- and sub-microcrystalline structures in IN718 and EP962 Ni-based alloys that are typically strengthened by phases, such as γ′ and γ″+δ. The investigation deemed that sub-microcrystalline structures can provide both a low temperature and the level of flow stress. Although a large number of studies have been reported, more literatures focus on Inconel 718 alloy. Among them, Smith and Flower studied Inconel 718 obtained a excellent elongation about 750% but the temperature was up to 954 °C and the initial strain rate at <1 × 10^−4^ s^−1^ [23]. Han et al. [24] reported that Inconel 718 alloy exhibited excellent elongation of 467% in the superplastic tensile test at 1000 °C and a strain rate of 1.14 × 10^−4^ s^−1^. The slip of the grain boundary or the continuous accumulation of the dislocation are the dominant mechanism. The growth of grain was observed after 980 °C, because the dissolution of *δ* phase at high temperature decreased the resistance of the migration of the grain boundary [24]. Huang et al. [25] studied the microstructural evolution of Inconel 718 alloy during the superplasticity and concluded that the discontinuous dynamic recrystallization (DDRX) is the main deformation mechanism for the superplasticity. Huang and Blackwell [26] carried out the optimum deformation condition for IN718 sheet at 965 °C with a strain rate of 10^−4^ s^−1^ and observed the grain size reduction. However, this study indicated that the grain reduction caused by amounts of low angle boundaries rather than the dynamic recrystallization (DRX), while no direct evidence was able to support their opinion. Furthermore, other nickel-based superalloys, such as PM IN-100 alloys [26], was also studied. Kikuchi et al. [26] reported that the superplastic deformation behavior of PM IN-100 alloys was investigated and the value of *m* obtained 0.6 at 1075 °C in the strain rate range of 10^−3^ s^−1^. The investigation deemed that PM IN-100 should be deformed at higher temperatures but below the γ′ solvus due to grain growth occurs and the microstructures with fine grains are not maintained caused by the dissolution of γ′ phase [26]. Meanwhile, grain refinement occurs due to dynamic recrystallization but no direct microstructural characterization in this study.

A number of investigations on the superplastic behavior of nickel-base superalloys have been reported. However, in the above investigations, the superplastic deformation in hard-to-deformed superalloys was not reported to be any systematic study, which is severely limited despite its significance for application. GH4151 was developed as a cast & wrought (C&W) nickel-based superalloy for disc superalloys, which is a typical hard-to-deformed superalloy due to high volume fractions of γ′ precipitation up to 55% and at a service temperature up to 800 °C. The volume fractions of γ′ precipitation is close to the limit of hot workability of C&W superalloys. Although the GH4151 alloy exhibits good application prospect due to the excellent mechanical properties, especially from 750 °C to 800 °C for turbine discs, cause by high value of alloying elements. The GH4151 alloy is very limited to a large number of applications as the result of aggravated hot workability. Therefore, the superplastic deformation provides a novel method to prepare the turbine disc for the GH4151 alloy. Meanwhile, for other hard-to-deformed superalloys, the superplastic deformation is a significant scheme in industrial applications. However, comparison with the Inconel 718, the investigation of superplastic deformation on hard-to-deformed superalloys, especially for GH4151 alloy, is rarely reported.

In the present investigation, the superplastic deformation of hot-extruded GH4151 alloy and its microstructure evolution have been systematically investigated. Further discussion paid more attention to the mechanism of superplastic deformation and DRX. The aim of this paper on the GH4151 alloy is to obtain the optimistic parameters of superplastic deformation and to identify its soft mechanism, which is significate for the superplastic forming of GH4151 turbine discs.

## 2. Materials and Experimental Procedure

A hot-extruded billet of GH4151 superalloy with a nominal diameter of 60 mm was used in the present study, and the chemical compositions are listed in Table 1. The thermodynamic equilibrium phase diagram was calculated by the phase calculation software, JMatPro, which predicts the phase composition of the GH4151 alloy versus temperature given the alloy composition according to the Table 1. As shown in Figure 1, the solvus temperature of gamma prime phase (γ’) was determined, and the high volume fraction of γ’ precipitation to 55% decreased with the temperature increasing and solved completely at roughly 1166.8 °C, which is the strengthening phase of the GH4151 superalloy. 

The microstructures after hot extruding are shown in Figure 2. Apparently, the grain size was uniform and ranged from 4 to 8 μm. Meanwhile, it is worth mentioning that the coarse gamma prime (γ’) phase, with 1 to 3 μm, pinned at the grain boundaries or triple-junction. Due to the pinning effect by coarse γ’ precipitates at grain boundaries, the grain structure could remain uniform and fine during the deformation blow the solvus temperature of γ’ precipitation. In addition, the coarse γ’ precipitates at grain boundaries is preferential nucleation sites for recrystallization, which is caused by the locally higher dislocation density and the resulting orientation gradient [27].

The superplastic tensile specimens, with a gage diameter of approximately 5 mm and gage length of about 25 mm as shown in Figure 3, were machined from the hot-extruded billet of GH4151 superalloy with a nominal diameter of 60 mm. The superplastic tensile tests were performed at 1060 °C, 1080 °C and 1100 °C with the initial strain rate of 10^−4^ s^−1^, 5 × 10^−4^ s^−1^ and 10^−3^ s^−1^ using the Instron 5982 tensile machine with a constant cross-head speed. The tensile machine was equipped with an electrical resistance furnace and an axial extensometer. The electrical resistance furnace with the temperature was controlled in three zones (an error less than 3 °C), and the tensile specimens were held at the deformation temperatures for 20 min prior to deformation. The tensile specimens were water quenched immediately after the tensile tests in order to freeze the deformed microstructure.

The deformed microstructure was characterized by optical microscope (OM), scanning electron microscope (SEM), electron backscatter diffraction (EBSD) and transmission electron microscope (TEM). The metallographic samples were achieved by mirror polishing followed by chemical etch dissolving. Cavities were observed in the as polished state. The grain structures were photographed using a Leica DMR (Wetzlar, Germany). The specimens for EBSD investigation was prepared by mirror polishing and followed by electropolished in 20% H_2_SO_4_ and 80% CH_3_OH at 20–30 V for 5–10 s at room temperature. The EBSD analysis was characterized by FEI Quanta 650 FEG-SEM (FEI Corporation, Hillsboro, OR, USA) and equipped with HKL Channel 5 software (Oxford Instruments plc, Abingdon, UK) operating at 20 KV. The EBSD data post-processing used in the EDAX-TSL OIM software (6.2.0 x86 version, EDAX Inc., Draper, UT, USA). The orientation image microscopy (OIM) maps, misorientation angle distributions of grains and grain size distribution could be calculated. The specimens for the observation of γ’/γ microstructure were prepared by mechanical polish and electrolytic etching in 9 g CrO_3_ + 90 mL H_3_PO_4_ + 30 mL C_2_H_5_OH at 5 V for 5–7 s. The γ’/γ microstructure were characterized using the secondary electron mode (SEI) of a JEOL JSM™ SU8000 field emission gun scanning electron microscope (FEG-SEM, JEOL Ltd., Tokyo, Japan) operating in at 3 KV.

Later, the foils for the transmission electron microscope (TEM) were cut from the tensile specimens close to the fracture surfaces. After low stress mechanical polishing to ~30 μm thickness, thin foils were prepared by ion milling, and the foils were observed using a JEM-2100F field emission gun transmission electron microscope (JEOL Ltd., Tokyo, Japan) operating at 200 kV.

## 3. Results and Discussion

### 3.1. Superplastic Behavior

The superplastic tensile stress–strain curves and tensile failure specimens at different conditions are shown in Figure 4. Obviously, the GH4151 alloy exhibited the excellent superplastic ductility and its elongation increased as the deformation temperature elevated from 1060 °C to 1100 °C. As shown in Figure 4a,c,e, the flow stress of all curves raised sharply to the peak values and then decreased slowly to the state-stage flow with the increasing strain until failure. The typical characteristic of flow curves indicated the occurrence of DRX during the dynamic softening process [28]. In addition, the values of peak stress ranged from 32–40 MPa, 22–26 MPa and 16–19 MPa corresponding to the deformation temperature of 1060 °C, 1080 °C and 1100 °C. It is indicated that the flow stress depends on the deformation temperature, and the values of peak stress decreased with the deformation temperature increasing. This is due to that the process of superplastic deformation is controlled by atomic diffusion [1], while the diffusion rate is positively correlated with temperature. Meanwhile, the effect of deformation temperature on the elongation to failure, at a different initial strain rate, is illustrated in Figure 5. With the deformation temperature increasing from 1060 °C to 1100 °C, the tensile ductility increased ranging from 280% to 760.4%. Obviously, the elongation to failure increased with the elevating temperature because of the atom diffusion rate increasing, which was consistent with diffusional creep. Moreover, it is noticeable that its elongation exceeded 280% under all experimental conditions and peaked the maximum value of 760.4% with 10^−3^ s^−1^ at 1080 °C.

The coefficient of strain rate sensitivity, *m*, can be expressed as function of logarithmic values of stress σ and strain rate $\dot{\varepsilon }$ shown in Figure 6b as [1,10,29]:(1)$$m={\frac{\partial \left(\ln\sigma \right)}{\partial \left(\ln\dot{\varepsilon }\right)}|}_{T}.$$

Generally, the materials exhibited the excellent superplasticity when the value of *m* exceeded 0.5. As shown in Figure 6b, the value of *m* exceeded 0.5 and increased from 0.52 to 0.73 with temperature under all experimental conditions and obtained the maximum value of 0.73 at 1080 °C. Apparently, GH4151 alloy revealed the optimal superplastic ductility under the experimental conditions.

The activation energy for superplastic deformation, *Q*, calculated by Arrhenius plots between ln $\dot{\varepsilon }$ and 1/T shown in in Figure 6a as [3,30]:(2)$$Q=\frac{1}{m}R{\frac{\partial \ln\sigma }{\partial \left(1/T\right)}|}_{\dot{\varepsilon }},$$
where *m* is the coefficient of strain rate sensitivity, *R* is the gas constant and *T* is the deformation temperature. From the slope of Figure 6a, the value of *Q* ranged from the 192 kJ/mol, 258 kJ/mol and 342 kJ/mol corresponding with strain rate of 1 × 10^−4^ s^−1^, 5 × 10^−4^ s^−1^ and 1 × 10^−3^ s^−1^, respectively. To our knowledge, the activation energy for superplastic deformation of GH4151 alloy is not reported. However, the value of *Q* on Inconel 718 is widely investigation. The *Q* of superplastic deformation with 1 × 10^−4^ s^−1^, 192 kJ/mol, is close to that for the grain boundary diffusion (107–198 kJ mol^−1^) [3]. However, the *Q* of superplastic deformation with 5 × 10^−4^ s^−1^ (258 kJ/mol) and 1 × 10^−3^ s^−1^ (342 kJ/mol) coincided better with that of volume self-diffusion energies (257–357 kJ/mol) [29]. Therefore, the superplastic deformation with 1 × 10^−4^ s^−1^ in this experiment is mainly deem as grain boundary sliding controlled by grain boundary diffusion. However, with higher stain rate of 5 × 10^−4^ s^−1^ and 1 × 10^−3^ s^−1^, the superplastic deformation is considered to be grain boundary sliding controlled by lattice diffusion.

### 3.2. Fracture Characteristics

After superplastic deformation to failure, no significant necking occurred at the superplastic fracture and the gauge sections of tensile specimens became very thin. Therefore, in the present study, fracture and cavities were observed in the as polished state in longitudinal plant. The cavitates were found to varying extents depending on the test conditions. Near the grip labeled as A, B and C respectively in Figure 7e, as illustrated in Figure 7a–c, the number of cavitation closing to the grip was found first to increase and then to decrease with temperature increasing, which is mainly nucleated at the niobium carbide particles [17]. Meanwhile, the size of cavitation became larger. However, near the fracture tip labeled as G, H and I respectively in Figure 7e, as shown in Figure 7g–i, the amount of cavitation was substantially increased. It is worth noting that cavities grew and interlinked along the force axis with the tensile ductility increasing. Whereas the cavities were non-uniformly distributed especially at 1080 °C (Figure 7h) and 1100 °C (Figure 7i) with 1 × 10^−3^ s^−1^. Moreover, larger cavities were occurred near the fracture, which may lead to the tensile specimens to failure. In addition, the cavitation of interrupted specimens was illustrated in Figure 7d,f at the position A and B in Figure 7e, which were interrupted at the tensile ductility of 61.8% and 350.5% under the test condition at 1080 °C with 1 × 10^−3^ s^−1^, respectively. Apparently, both of the amounts and size of cavitation increased with the increase of tensile ductility. Similarly, the neighborhood of the niobium carbide particles is the preferential nucleation site for cavitation, and the cavities were connected by interaction with each other as the strain increases. The low ductility was caused by cavitation forming and reaching catastrophic values.

### 3.3. Deformation Microstructure

Figure 8 illustrates the microstructural evolution near the fracture after failure with strain rate of 1 × 10^−3^ s^−1^ at deformation temperatures of 1060 °C, 1080 °C and 1100 °C, respectively. Figure 8c,e were compared with Figure 8a, and it is clearly found that the final grain size increases with the deformation temperature increasing. The results were attributed from both the temperature increasing and the superplastic deformation. On the one hand, the migration of the grain boundary is a thermal activation process, the rate of grain boundary increases with increasing temperature. On the other hand, previous investigation reported that the dynamic growth of grain is accelerated by superplastic deformation and possible due to the stress [17]. In addition, compared with the initial microstructure in Figure 2a, the grain refinement occurs during superplastic deformation due to DRX, which is also reported by previous studies in superplastic materials.

### 3.4. Dynamic Recrystallization 

The typical characteristic of stress–strain curves illustrated that the stress rose sharply to the peak values followed by dynamic softening slowly to the state stage.

The grain refinement occurs during superplastic deformation. Both of these indicate the occurrence of DRX on GH4151 alloy during superplastic deformation. To investigate the microstructural evolution during superplastic process, the superplastic tensile tests at 1080 °C and 1 × 10^−3^ s^−1^ were interrupted at the elongation of 61.8% and 350.5%, respectively. Figure 9 presents the OIM maps and grain misorientation angle distributions of the deformed specimens under the ductility of 61.8%, 350.5% and 760.4% at 1080 °C with strain rate of 1 × 10^−3^ s^−1^. As shown in Figure 9a, the elongated grain was clearly observed in the deformed specimens interrupted at the ductility of 61.8%. The fraction of low angle grain boundaries (<15°, LAGBs) is high and the average misorientation angle is 17.25°, as illustrated in Figure 9b. Similarly, interrupted at the ductility of 350.5%, the LAGBs decreased slightly relative to that of the specimens interrupted at 61.8%. However, with the specimens deforming to failure at the ductility of 760.4%, the LAGBs reduced and the high angle grain boundaries (≥15°, HAGBs) increased distinctly. This observation confirmed the occurrence of DRX during the superplastic deformation [25].

In order to further investigate the mechanism of DRX during superplastic deformation, TEM observations were carried out. Figure 10 illustrates the TEM microstructure of the specimens interrupted at the elongation of 61.8% at 1080 °C with strain rate of 1 × 10^−3^ s^−1^. As shown in Figure 10a, the random dislocations were observed and both of dislocation network and array were formed. With the strain increasing, the random dislocations climbed, slipped and recombined into dislocation array, grain boundaries and the triple junction, as illustrated in Figure 10a,b. The dislocation pile-up was observed in Figure 10b, which was caused by deformation coordination at the triple junction. From the TEM observation, the dislocation motion, such as the climbing and slipping, was the important deformation substructures and attributed to the ductility of superplastic deformation in the GH4151 alloy. 

Generally, based on the mechanism of occurrence, DRX can be classified into two kinds: continuous dynamic recrystallization (CDRX), discontinuous dynamic recrystallization (DDRX) and geometric dynamic recrystallization (GDRX) [31]. On the contrary to CDRX, DDRX will be accompanied by nucleation and recrystallization grain growth. However, the GDRX usually occurs in pure metals, which is found at high deformation temperature with low strain rate and large strains [32]. In present study, it is particularly noticeable that the subgrains bulge formed at the original grain boundaries, as presented in the Figure 10c,d. The observation is a typical characteristic of DDRX, which was reported in previous investigations [33,34,35,36]. Furthermore, the new grains are observed to nucleate in the triple junction, as shown in Figure 10e,f. The triple junction is deemed as the preferential nucleating sites for DRX, which is because the grain rotation, grain boundaries migration and sliding would prompt the triple junction to DRX nucleation [37]. In addition, it is known that the process of superplastic deformation associated with the propagation, slip, annihilation and recombination of dislocation. Moreover, the dislocation would be absorbed or annihilated during the DRX nucleation, which resulted in the density of dislocations decreasing, as presented in Figure 10.

Figure 11 illustrated the TEM microstructure of the specimens interrupted at the elongation of 350.5% at 1080 °C with strain rate of 1 × 10^−3^ s^−1^. As shown in Figure 11a, with the tensile ductility increasing up to 350.5%, the DRX grain nucleated at the original grain boundary, and the migration of DRX grain boundary was observed. The nucleation of DRX at the triple junction found in Figure 11b was observed in the specimens interrupted at the elongation of 61.8%. Furthermore, it is worth noting that the dislocation cell formed at grain boundary, as exhibited in Figure 11c. Dislocations moved towards the LAGB to increase its misorientation, then the LAGB turn into HAGB when its misorientation exceeds 15°. Distinct from DDRX, the type of DRX process did not involve nucleation, which is consistent with the DDRX mechanism. In addition, it is particularly noticeable that some bands of grain boundaries, which is confirmed that the grain boundary sliding occurred for the GH4151 alloy during the superplastic deformation [21]. From TEM observation, above all, it can be identified that DDRX is the dominant DRX mechanism and CDRX is merely an assistant mechanism of DRX for GH4151 alloy during the superplastic deformation. 

Here, it is necessary to discuss the role of dynamic recrystallization in superplastic deformation. Generally, grain boundary sliding was considered as the dominant deformation mechanism of superplasticity. In this paper, as shown in Figure 10a,b, the motion of the intergranular dislocations was observed. However, the dislocations were rarely found during the superplastic deformation, as illustrated in Figure 10c–f and Figure 11. For the alloys with low stacking fault energy, the dislocation multiplication and movement stored the energy of deformation during the superplastic deformation. Meanwhile, the dislocation density increased to the critical level of DRX with the increase of superplastic deformation, which induced the occurrence of DRX. As a result, the excellent superplasticity was a benefit from the grain refinement. Therefore, the main contributions of DRX on superplasticity elongation were derived from its grain refinement process.

## 4. Conclusions

For the hard-to-deformed superalloy of GH4151, the superplastic behavior has been investigated for the first time in this investigation. The following results were obtained. (1)The stress–strain curves demonstrated that the flow stress increased rapidly and followed by dynamic softening. The values of peak stress ranged from 32–40 MPa, 22–26 MPa and 16–19 MPa corresponding to the deformation temperatures of 1060 °C, 1080 °C and 1100 °C, respectively. The value of *m* exceeded 0.5 and varied from 0.52 to 0.73. The maximum ductility of 760.4% was obtained at a strain rate of 1 × 10^−3^ s^−1^ and 1080 °C.(2)The activation energy for superplastic deformation were 192 kJ/mol, 258 kJ/mol and 342 kJ/mol corresponding to strain rates of 1 × 10^−4^ s^−1^, 5 × 10^−4^ s^−1^ and 1 × 10^−3^ s^−1^, respectively. The deformation mechanism of superplasticity was mainly deemed as grain boundary sliding controlled by grain boundary diffusion at the strain rate of 1 × 10^−4^ s^−1^, while it was considered to be grain boundary sliding controlled by lattice diffusion with higher stain rates of 5 × 10^−4^ s^−1^ and 1 × 10^−3^ s^−1^.(3)Grain growth and cavitation were observed during superplastic deformation. The grain size increased obviously after superplastic deformation to failure at 1080 °C and 1100 °C. Based on the microstructural observation, DRX occurred during the superplastic deformation. DDRX was the dominant DRX mechanism and CDRX was merely an assistant mechanism of DRX for the GH4151 alloy during the superplastic deformation. 

## Figures and Tables

**Figure 1 materials-12-03667-f001:**
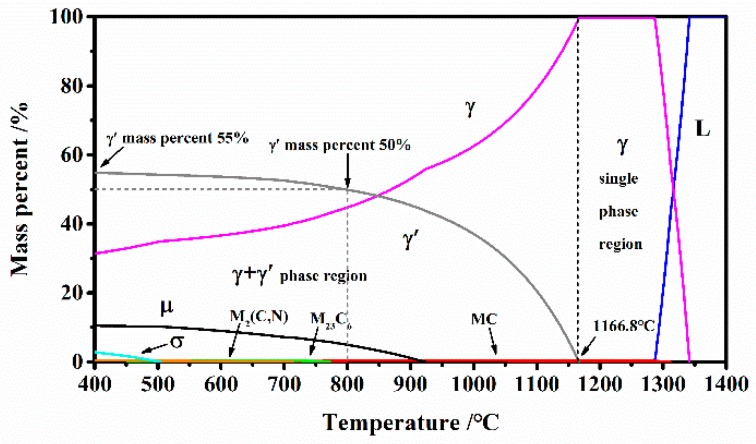
Thermodynamic phase diagram calculated by JMatPro software.

**Figure 2 materials-12-03667-f002:**
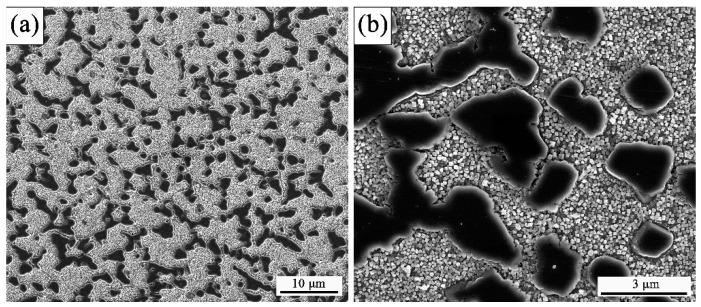
(**a**,**b**) The initial microstructure of the hot-extruded GH4151 alloy.

**Figure 3 materials-12-03667-f003:**
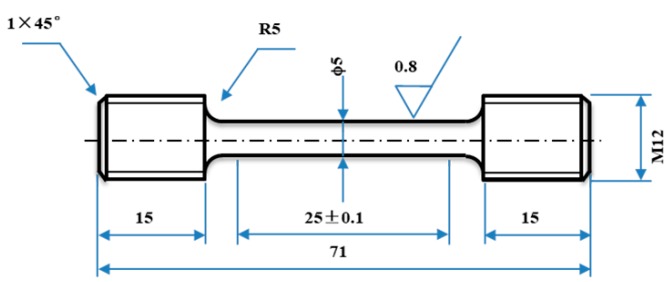
Dimension of tensile specimen.

**Figure 4 materials-12-03667-f004:**
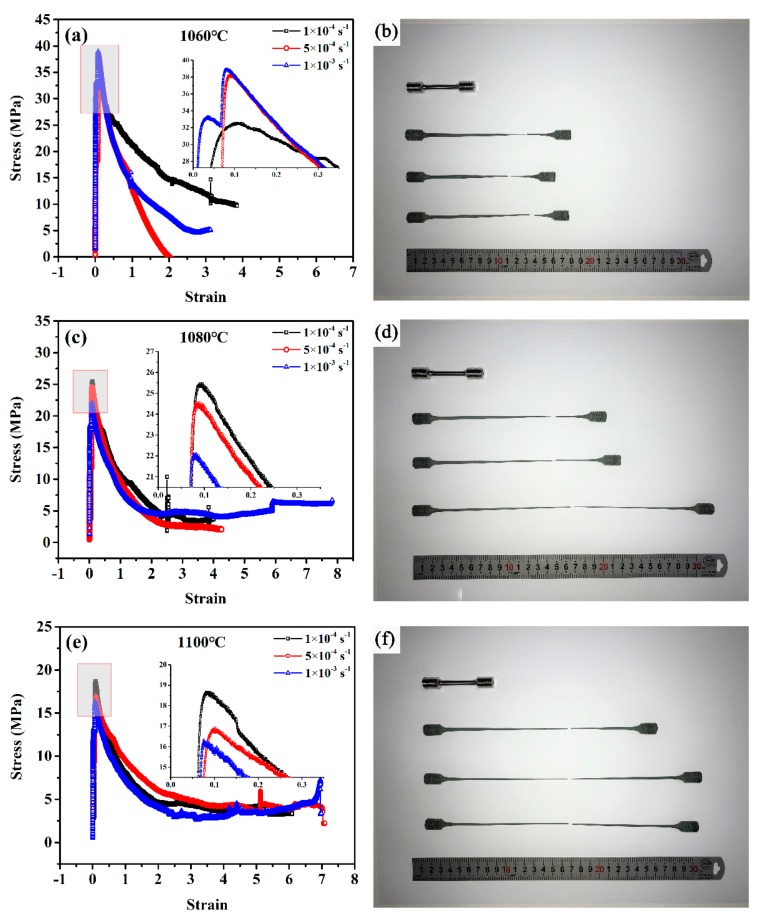
The stress–strain curve and tensile failure specimens at 1060 °C (**a**,**b**), 1080 °C (**c**,**d**) and 1100 °C (**e**,**f**) with the initial strain rate of 10^−4^ s^−1^, 5 × 10^−4^ s^−1^ and 10^−3^ s^−1^, respectively.

**Figure 5 materials-12-03667-f005:**
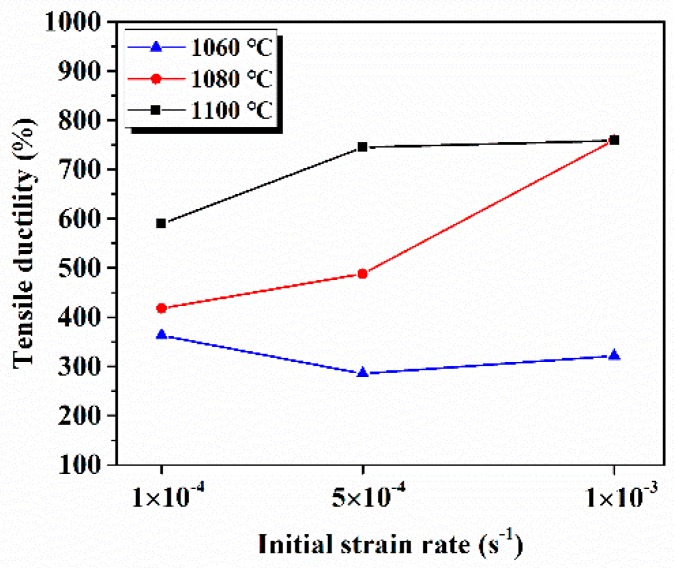
The superplastic tensile ductility to failure as a function of various initial strain rates at test temperatures.

**Figure 6 materials-12-03667-f006:**
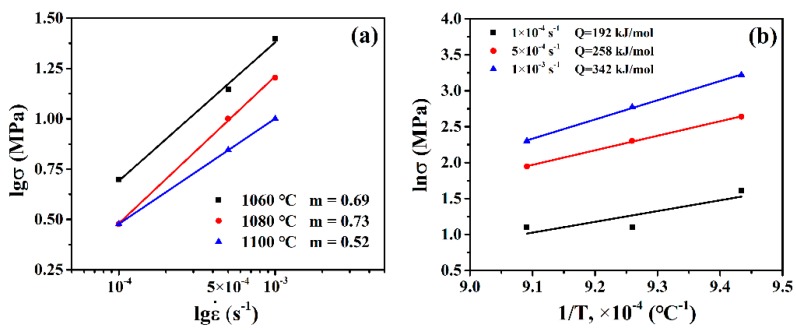
(**a**) Flow stress as a function of strain rate at various temperatures and (**b**) ln s as a function of reciprocal temperature.

**Figure 7 materials-12-03667-f007:**
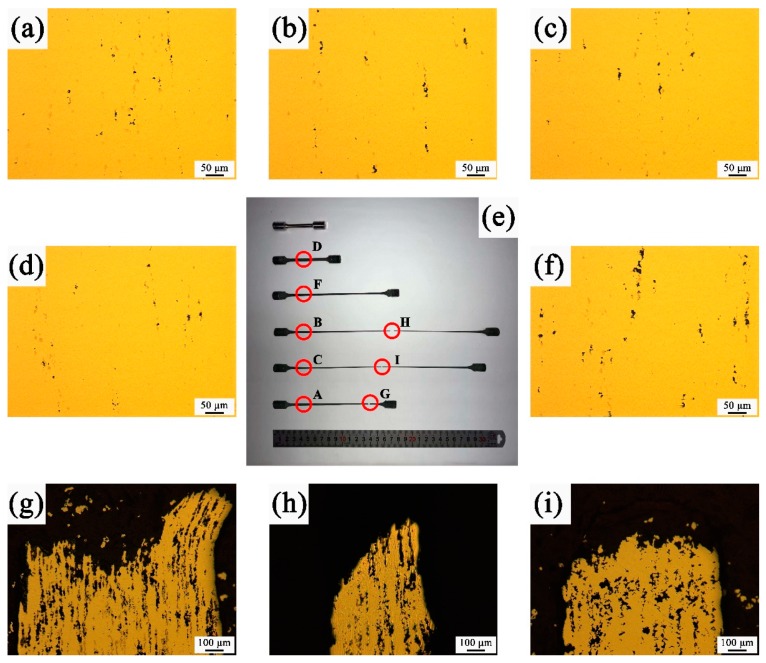
Variation in the amount of cavitation and fracture morphologies in the Tensile specimens (**e**) to failure with the strain rate of 1 × 10^−3^ s^−1^ at 1060 °C (**a**,**g**); 1080 °C (**b**,**h**); 1100 °C (**c**,**i**) and interrupted at 1080 °C with strain rate of 1 × 10^−3^ s^−1^ at different ductility, respectively: (**d**) 61.8% and (**f**) 350.5%.

**Figure 8 materials-12-03667-f008:**
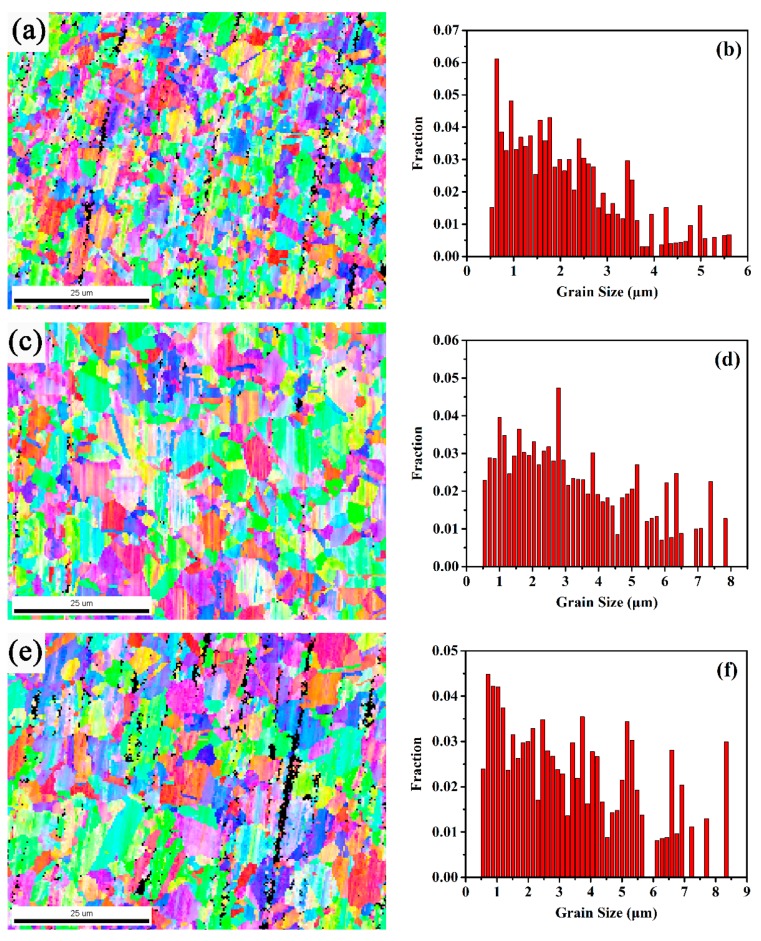
Orientation image microscopy (OIM) maps and grain size distribution of the specimens to failure with strain rate of 1 × 10^−3^ s^−1^ at different temperature, respectively: (**a**,**b**) 1060 °C, (**c**,**d**) 1080 °C and (**e**,**f**) 1100 °C.

**Figure 9 materials-12-03667-f009:**
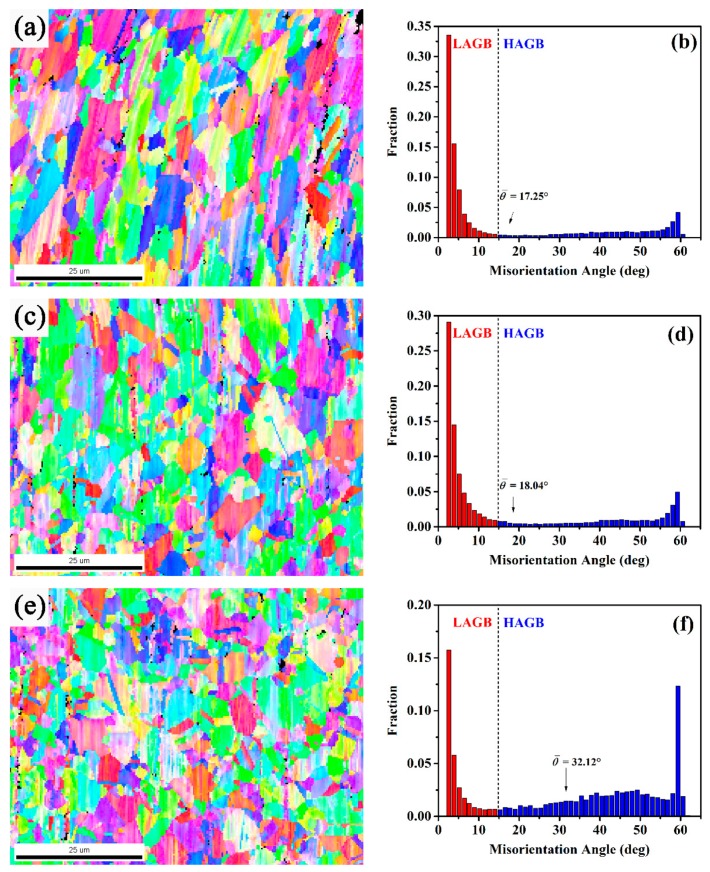
OIM maps and grain misorientation angle distributions of the deformed specimens at 1080 °C with strain rate of 1 × 10^−3^ s^−1^ at different ductility, respectively: (**a**,**b**) 61.8%, (**c**,**d**) 350.5% and (**e**,**f**) 760.4%.

**Figure 10 materials-12-03667-f010:**
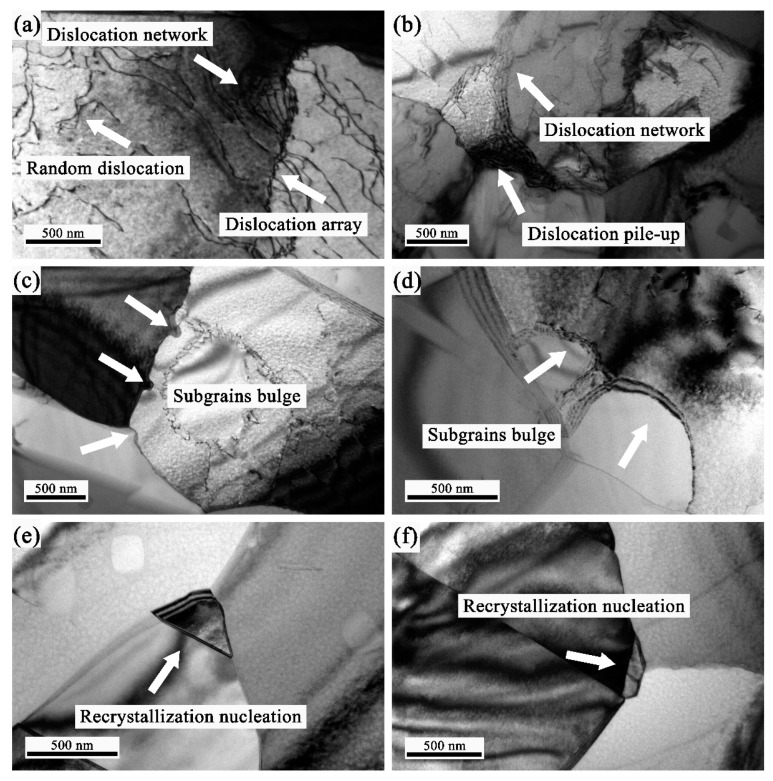
Typical TEM images of the alloy deformed to the ductility of 61.8% at 1080 °C with a strain rate of 1 × 10^−3^ s^−1^: (**a**,**b**) the morphology of dislocations, (**c**,**d**) subgrains bulge, (**e**,**f**) recrystallization nucleation.

**Figure 11 materials-12-03667-f011:**
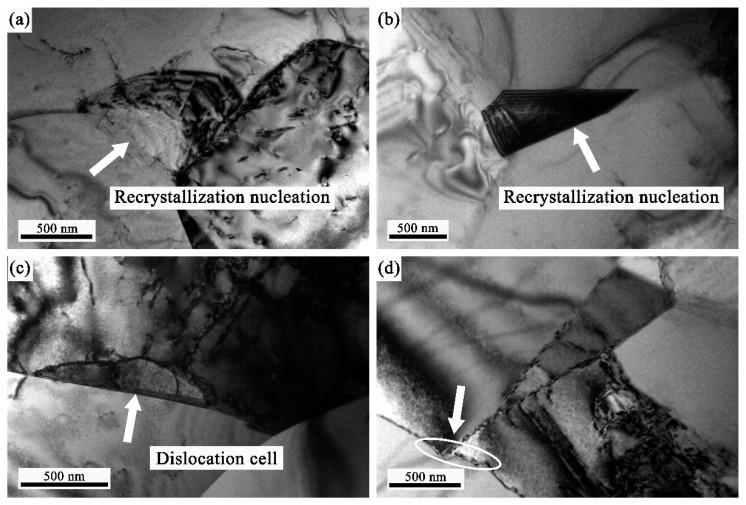
Typical TEM images of the alloy deformed to the ductility of 350.5% at 1080 °C with a strain rate of 1 × 10^−3^ s^−1^: (**a**,**b**) recrystallization nucleation, (**c**) the dislocation cell formed at grain boundary, (**d**) bands of grain boundaries.

**Table 1 materials-12-03667-t001:** Chemical compositions of the GH4151 superalloy (major elements).

Elements	Ni	C	Cr	Co	Mo	W	Al	Ti	Nb	V
**wt./%**	Balanced	0.07	10.0	15.0	4.5	3.0	3.6	2.8	3.4	0.5

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
