# Peer review of "Superplastic Deformation and Dynamic Recrystallization of a Novel Disc Superalloy GH4151"

_materials, 2019, doi:10.3390/ma12223667_

Round 1
Reviewer 1 Report
Review
The submitted manuscript describes a superplasticity of nickel-based superalloy. The authors outline in several places of manuscript that superplasticity of GH4151 was reported for the first time, what literally is true but superplasticity of multiple other, very similar nickel-based superalloys was already described in numerous original and review scientific papers. Thus, the presented data are new but not novel and the general findings about superplasticity are expected and typical. Nevertheless, in opinion of reviewer, they manuscript still can be interesting to readers as presented data can be useful in further research and possibly can find specific applications. Because manuscript possesses some minor errors (described below), the reviewer cannot recommend publication of the manuscript in the current form and suggests revision.
Comments
The reference to the basic review paper on nickel-based superalloys: “Kaibyshev O.A. (1992) Superplasticity of Nickel-based Superalloys. In: Superplasticity of Alloys, Intermetallides and Ceramics. Springer, Berlin, Heidelberg DOI https://doi.org/10.1007/978-3-642-84673-1_9” should be added to manuscript. This work, despite the year of publishing, is still the most comprehensive on the described topic. Beside above the following references to closely related papers should be added to manuscript: S. X. McFadden, R. S. Mishra, R. Z. Valiev, A. P. Zhilyaev & A. K. Mukherjee; Nature volume 398, pages684–686 (1999); Valitov, V. & Bewlay, B. & Mukhtarov, Shamil & Kaibyshev, O.A. & Gigliotti, Michael. (2000). Superplasticity of Nickel-Based Alloys with Micro- and Sub-Microcrystalline Structures. Materials Science Forum - MATER SCI FORUM. 601. 10.1557/PROC-601-43.
The observed superplasticity occurs at relatively high temperatures (above 1000 deg. C) what is common phenomenon for nickel-based superalloys (as outlined in above cited references). It should be mentioned in manuscript.
The authors select too small number of temperature values. It is especially visible in observed tensile ductility as a function of initial strain rate. This function is monotonic at 1080, 1100 deg C and is not monotonic at 1060 deg C. At which temperature there is a change of not monotonic to monotonic function? Additionally, at larger temperatures the initial strain rate affects weaker tensile ductility. It is important, is there a temperature (before melting) at which initial strain rate does not affect tensile ductility? Authors should study the wider temperature range, with smaller gradation of temperature periods (2 deg. or 5 deg. instead of 20 deg.) to explain these questions. Additionally, authors should study the wider range of initial stress rates, with smaller gradation of initial stress rates periods.
The test should be made at least in duplicate to make sure that presented results are repeatable.
The microstructure typically changes after stress. Authors study and correctly present local changes of microstructure, but registration of micrographs at smaller magnification (10 and 3 micrometres) will be beneficial to readers, as it allows comparison with global microstructure of original sample (as presented in figure 1).
Reviewer 2 Report
The authors present an experimental study on the superplastic deformation of GH4151 alloy using tensile tests under different temperatures and strain rates; a systematic microstructural examination was also performed to understand the failure and dynamic recrystallization mechanisms.
I accept the publication of the paper; however, moderate English changes are required. For example: line 32: "fabrication varied" should be "fabrication of varied" or "fabricating varied"; line 33: "numbers investigations" should be "numerous investigations".
Author Response
Dear Ms. Freya Dong and Reviewer,
Thank you for your kind consideration of our manuscript “ Superplastic Deformation and Dynamic Recrystallization of a Novel Disc Superalloy GH4151” (Manuscript ID: materials-627767) and for sending us the valuable comments.
According to your suggestion and the referees’ comments, we have carefully revised the above mentioned manuscript and hope now it is suitable for publication in Materials.
Below please find the point to point response to the review comments. The detailed change/modification has been highlighted in the revised manuscript.
Point 1: I accept the publication of the paper; however, moderate English changes are required. For example: line 32: "fabrication varied" should be "fabrication of varied" or "fabricating varied"; line 33: "numbers investigations" should be "numerous investigations".
Response 1: Thank you for your suggestion. The related mistakes have been corrected. As suggested, the manuscript has been carefully rechecked. The modifications are presented in the modified version of manuscript, which are marked in color of red. The detailed modifications are shown as follows:
line 32: "fabrication varied" should be "fabrication of varied" or "fabricating varied";
line 33: "numbers investigations" should be "numerous investigations".
line 154:"all of the experimental conditions" should be "all experimental conditions".
line 166:"all of the experimental conditions" should be "all experimental conditions".
Thank you again and we are looking forward to hearing from you again soon. Thank you!
Best regards,
Chonglin Jia
Doctor
Science and Technology on Advanced High Temperature Structural Materials Laboratory
Beijing Institute of Aeronautical Materials
Beijing 100094
P.R. China
E-mail: [email protected]
Round 2
Reviewer 1 Report
In revised manuscript authors introduced alterations according to the some reviewer comments. In case of comments not used in preparation of revision, authors explain the reason their omitting. Generally it should be considered as a difference in opinions between scientists, not as errors of authors, however finding response to two question arising during reading of manuscript (1. At which temperature there is a change of not monotonic to monotonic function of tensile ductility against initial strain rate?; 2. Is there a temperature (before melting) at which initial strain rate does not affect tensile ductility?) will be very valuable and beneficial for manuscript and its readers. It is pity that authors divide results of one narrow research topic into two manuscripts submitted to different journals (as it is described in response no. 5), however the reviewer is aware that it is currently hard to combine both manuscripts into one. Considering above the reviewer recommend publishing of manuscript in the Materials.